# Rheology by Design: A Regulatory Tutorial for Analytical Method Validation

**DOI:** 10.3390/pharmaceutics12090820

**Published:** 2020-08-28

**Authors:** Ana Simões, Margarida Miranda, Catarina Cardoso, Francisco Veiga, Carla Vitorino

**Affiliations:** 1Faculty of Pharmacy, University of Coimbra, Pólo das Ciências da Saúde, Azinhaga de Santa Comba, 3000-548 Coimbra, Portugal; simoesana88@gmail.com (A.S.); mimiranda@live.com.pt (M.M.); fveiga@ff.uc.pt (F.V.); 2Associated Laboratory for Green Chemistry of the Network of Chemistry and Technology (LAQV. REQUIMTE) Group of Pharmaceutical Technology, Faculty of Pharmacy, University of Coimbra, Pólo das Ciências da Saúde, Azinhaga de Santa Comba, 3000-548 Coimbra, Portugal; 3Coimbra Chemistry Center, Department of Chemistry, University of Coimbra, Rua Larga, 3004-535 Coimbra, Portugal; 4Laboratórios Basi, Mortágua, Parque Industrial Manuel Lourenço Ferreira, lote 15, 3450-232 Mortágua, Portugal; catarina.cardoso@basi.pt; 5Centre for Neurosciences and Cell Biology (CNC), University of Coimbra, Rua Larga, Faculty of Medicine, Pólo I, 1st floor, 3004-504 Coimbra, Portugal

**Keywords:** rheology, method validation, equipment qualification, topical dosage forms, regulatory requirements

## Abstract

The increasing demand for product and process understanding as an active pursuit in the quality guideline Q8 and, more recently, on the draft guideline on quality and equivalence of topical products, has unveiled the tremendous potential of rheology methods as a tool for microstructure characterization of topical semisolid dosage forms. Accordingly, procedure standardization is a dire need. This work aimed at developing and validating a methodology tutorial for rheology analysis. A 1% hydrocortisone cream was used as model cream formulation. Through a risk assessment analysis, the impact of selected critical method variables (geometry, temperature and application mode) was estimated in a broad range of rheological critical analytical attributes—zero-shear viscosity, upper-shear thinning viscosity, lower-shear thinning viscosity, infinite-shear viscosity, rotational yield point, thixotropic relative area, linear viscoelastic region, oscillatory yield point, storage modulus, loss modulus, and loss tangent. The proposed validation of the approach included the rheometer qualification, followed by the validation of numerous operational critical parameters regarding a rheology profile acquisition. The thixotropic relative area, oscillatory yield point, flow point and viscosity related endpoints proved to be highly sensitive and discriminatory parameters. This rationale provided a standard framework for the development of a reliable and robust rheology profile acquisition.

## 1. Introduction

Topical semisolid dosage forms include complex multiphase systems demanding a detailed rheological characterization, since these properties may meaningfully affect quality and performance [1,2]. Rheology regards the study of the material flow and deformation behaviour and may be measured by applying an external force (shear-induced deformation) to a sample [3]. Allied to formulation viscosity, elasticity and plasticity, rheological behaviour may impact product manufacturing, appearance, packaging, long-term stability, dispensing, sensory properties and the in vivo performance [4,5]. Therefore, rheology assessment proves to be an useful quality and stability indicator, revealing predictive information concerning batch variability, product aesthetic properties, therapeutic effectiveness and patient compliance [2,6,7]. An enhanced understanding and control of rheology parameters is the basis for the sustainable development of new or abbreviated drug applications, meeting stakeholders’ expectations [8].

Topical semisolid dosage forms predominantly exhibit a non-newtonian behaviour, since a higher shear rate induces a viscosity decrease, which enables an easier skin application [9]. As such, a given critical stress value (yield stress) is required for the formulation to start to flow. Below this point, the products majorly present elastic properties; on the contrary, above this endpoint, the material predominantly displays a plastic flow [10]. 

Likewise, there are a plethora of rheology attributes which directly influence topical product microstructure and, consequently, impact several aspects. For instance, the formulation spreadability and bioadhesion to the skin are highly affected by viscoelastic properties. As patients directly apply topical formulations on their skin, these sensorial attributes are of outmost importance to assure patient acceptability and, therefore, treatment compliance [10,11,12,13]. 

Stability and physical appearance also depending on rheological features. A detailed rheological characterization provides valuable insight on why products may settle or separate over shelf life. Furthermore, this tool can determine if there is a significant impact on product microstructure whenever dispensed from a packaging tube/dosing pump [10]. Biopharmaceutical characteristics, such as drug release and permeation are also reliant on the formulation rheological profile [13,14]. For all the above reasons, rheology behaviour is a key quality attribute within a target product profile of semisolid formulations [5,10,15]. 

Rheological characteristics are highly dependent on critical material attributes (CMAs) and critical process parameters (CPPs), therefore, a close rheological monitoring can be a useful tool to guide and shorten product development, as well as to assure product quality and reduce batch variations during manufacturing [7]. This is in line with the pharmaceutical industry growing need to gain process understanding and improve product quality. These are the underlying principles of Quality by Design (QbD). This new pharmaceutical regulatory concept is based on a systematic and risk-based approach, where the desired product quality profile is modulated through a detailed understanding of both raw materials and process parameters [11,16,17]. 

QbD methodology firstly involves the definition of the quality target product profile (QTPP) and critical quality attributes (CQAs), in which rheological characteristics should be a primary concern. Afterward, through a detailed risk analysis, both CMAs and CPPs should be clearly identified. With these parameters well established, design of experiments (DoEs) should be performed in order to finally establish the design space, as well as a viable control strategy [11,16,17]. This final step, is of outmost importance, since it warrants that the process is controlled and kept within the established design space [18]. Measurements during manufacturing with process analytical technologies (PAT) can be integrated as a part of a control strategy. Even though PAT is increasingly applied in solid dosage forms, its application in semisolid formulations is not yet seen as a common solution [19]. Nevertheless, several authors have been exploring the potential of rheology as a PAT tool. 

Qwist et al. have developed a pressure difference apparatus which can sample from the bulk intermediate/product stream and determine the storage modulus (G′) and the loss modulus (G″), through the frequency sweep test, as well as the flow curve [19]. Furthermore, Van Heugten and colleagues have evaluated the influence of filling temperature on an ointment yield stress. Based on this knowledge, the authors were able to establish an optimal filling viscosity range, which, in turn, enabled a successful filling operation with minimal weight variation and a product with the desired yield stress [20]. 

Once variability is understood, a more flexible regulatory approval can be attained if QbD principles are followed. As part of this strategy, the application of rheology as a PAT tool can be helpful to improve formulation and manufacturing capabilities, by reducing product variability and batch rejection [11,16]. 

As outlined in the draft guideline on the quality and equivalence of topical products, a patient-focused approach should be envisioned while developing a product [21,22]. Therefore, as previously mentioned, aspects such as patient acceptability, highly influenced by rheological attributes, should be primary concerns when developing a product. This is valid for an innovator product, but it is also highly relevant when addressing a generic product.

EMA draft guideline proposes, as an alternative to clinical endpoint studies, a modular framework for equivalence demonstration in topical generic products. Accordingly, for a product to apply, extended pharmaceutical equivalence criteria must be fulfilled: (i) qualitative, quantitative and microstructure sameness (Q1, Q2, Q3, respectively) towards the reference product; (ii) product performance (Q4) mainly supported by in vitro release testing; and, finally, (iii) if the test product regards a complex dosage form, equivalence regarding the efficacy profile should be supported through in vitro permeation or dermatopharmacokinetic studies [22,23]. In this context, microstructure equivalence demonstration is a cornerstone for bioequivalence assessment of topical generic products. 

There are multiple factors, broadly described in the literature, that influence microstructure, and wherein rheology attributes play an irrefutable role [5,6,23,24,25]. For this reason, EMA presents specific requirements concerning the rheological parameters that should be accomplished while describing the rheological behaviour of a give formulation [21]. These include: (i) a complete flow curve of shear stress (or viscosity) vs. shear rate; (ii) yield point values; (iii) linear viscoelastic response, (iv) storage and loss modulus vs. frequency/stress; and (v) thixotropic relative area. Even though FDA also requires the presentation of rheological endpoints, the list is not as exhaustive when comparing to the European agency. 

However, and despite the existence of several literature reports concerning the applicability and overall importance of rheology, there is a lack of understanding and standardization regarding formal validation procedures of such technique. Neither the parameters that define semisolid rheology profile, nor their acceptance limits have so far been defined in the literature. Furthermore, crucial rheology parameters are not included as routine analysis when releasing new batches [6]. In this context, a widespread validation applied to all semisolid dosage forms should be provided, safeguarding that the developed rheology measurement methods have suitable discriminatory abilities to determine formulation “sameness” and also, to detect formulation differences, which may affect clinical performance [26]. Moreover, a detailed rheology profile, with mandatory quality parameters, should also be available.

Aiming to standardize the rheological methodology, whether for assisting quality control or even a potential PAT tool, a comprehensive characterization of the rheometer operational parameters that could impact the rheology profile was carried out. To this end, the assumptions of the analytical quality by design (aQbD), including risk assessment applied to rank the impact of critical method variables (CMV) over critical analytical attributes (CAA), were considered to systematically validate the operational ranges of the rheometer, the experimental setup and the rheology measurement methods for the acquisition of a suitable rheology profile. 

Specific rotational and oscillatory measurements, alongside with data analysis were carried out considering all the relevant components of a conventional analytical validation, including precision, discriminatory power and robustness [27]. A 1% *w/w* hydrocortisone cream was used as a model product. Additional recommendations pointed out in the draft guideline on quality and equivalence of topical products were likewise addressed [21]. 

## 2. Materials and Methods

### 2.1. Materials

Micronized HC was kindly provided by Laboratórios Basi Indústria Farmacêutica S.A. (Mortágua, Portugal). Methyl parahydroxybenzoate and propyl parahydroxybenzoate were purchased from Alfa Aesar (Kandel, Germany). Kolliwax^®^ GMS II (glycerol monostearate), Kolliwax^®^ CA (cetyl alcohol), Kollicream^®^ IPM (isopropyl myristate), and Dexpanthenol Ph. Eur. were kindly provided by BASF SE (Ludwigshafen, Germany). Stearic acid was provided by Acorfarma distribuicion S.A. (Madrid, Spain). Triethanolamine was purchased from Panreac AppliChem (Darmstadt, Germany). Liquid paraffin was provided by LabChem Inc. (Zelienople, PA, USA). Glycerol was purchased from VWR Chemicals (Leuven, Belgium). Water was purified (Millipore^®^) and filtered through a 0.22 mm nylon filter before use. Viscosity reference standard RT5000 (Fungilab, Spain) was used for rheometer qualification studies. 

### 2.2. Methods

#### 2.2.1. Preparation of HC Cream Formulations

HC o/w cream formulations were conventionally prepared resorting to an Ultra-Turrax X 10/25 (Ystral GmbH, Dottingen, Germany) equipment. Both continuous and dispersed phases were separately prepared and heated to 70 °C [11,28]. Afterward, the active pharmaceutical ingredient was solubilized in the dispersed phase. Previous studies established the optimal experimental settings relating to rate, duration and temperature of the manufacturing process. After production, cream formulations were cooled down to room temperature. Batches of 0.5 Kg were considered. All samples were stored at 20–25 °C.

In order to document the discriminatory power of the proposed rheological analysis, three formulations were prepared considering different concentration of glycerol monostearate: 5% (F_5_), 10% (F_10_) and 20% (F_20_). This excipient bears a significant impact on product microstructure due to its thickening properties [11,28]. Please note that F_10_ was considered as the reference formulation. Moreover, a forth formulation was prepared with 10% of glycerol monostearate, but considering a different homogenization rate during manufacture. This formulation will be further addressed as a F_10_ negative control (F_10NC_). 

#### 2.2.2. Rheological Characterization

The rheological profile of all products was investigated using a HAAKE MARS 60 6000 (Thermo Scientific, Karlsruhe, Germany) equipped with a peltier system as temperature control unit. The data was evaluated using the Haake RheoWin Data Manager software (Thermo Scientific, Karlsruhe, Germany). For every analysis, a sample hood was used to minimize temperature fluctuations. 

Considering EMA recently published draft guideline on quality and equivalence of topical products, a complete rheological profile should include both rotational and oscillatory measurements [21]. 

Rotational tests are sample destructive. The information retrieved from these measurements enables the assessment of small periodic deformations, which affect structural breakdown and/or rearrangement. Moreover, with these tests the ability of a material to recover can also be studied.

Oscillatory tests regard amplitude and frequency sweep tests. Generally, these measurements, due to the decreased shear stress applied, can be considered as non-destructive, nevertheless, it should be pointed out that minor system perturbations can still occur during amplitude sweep tests. Oscillatory measurements aim to assess the material viscoelastic properties, while exposed to small-amplitude deformation forces [28]. The following sections detail the main outputs of both methodologies.

##### Rotational Measurements

Rotational tests were performed with a C35/2°/Ti cone geometry at 32 °C. Approximately 0.3 g of formulation were placed on a lower plate TMP35 using a positive displacement syringe. A pre-set gap of 0.1 mm was considered. 

A linear CS step test from 0.01 to 250 Pa was measured for 800 s, to trace the flow curve [ƞ = f(τ)]. To characterize the flow behaviour, the following responses, or critical analytical attributes (CAA), were determined: zero-shear viscosity (ƞ_0_), upper-shear thinning viscosity (ƞ_U_), lower-shear thinning viscosity (ƞ_L_), infinite-shear viscosity (ƞ∞) and yield point (τ_0.ROT_). 

To evaluate the thixotropic behaviour, a CR ramp test was performed with a shear rate from 0.01 to 300 s-1 and down again to 0.01, during 300 s^−1^ [τ = f($\dot{ɣ}$)]. From this analysis, the thixotropic relative area (SR) was calculated.

##### Oscillatory Measurements

The viscoelastic properties were investigated using a P35/Ti plate geometry at 32 °C. Approximately 1 g of the formulation was applied on a lower plate TMP35, using a syringe. An amplitude sweep test between 0.01 and 600 Pa at 1 Hz was firstly conducted to estimate the linear viscoelastic region (LVR) plateau, yield point (τ_0.OSC_) and flow point (τ_f_). Afterward, a frequency sweep analysis was conducted within the LVR plateau. The storage modulus (G′), loss modulus (G″) and loss tangent (tan δ) were calculated at 1 Hz.

#### 2.2.3. Rheological Method Validation

The present work aimed to establish a practical and straightforward approach concerning the validation of a rheological analysis. In this context, following a traditional validation procedure, precision and robustness, alongside with sensitivity, specificity, selectivity (discriminatory power) were determined [21,29]. Please note that linearity was not considered a relevant parameter for the rheological method validation, since there is no inherent linearity within the acquisition of a rheological profile.

##### Risk Assessment

According to prior knowledge, it was possible to extensively identify the analytical settings—CMV—which may pose a direct repercussion on rheological endpoints. To determine which of these parameters need to be further studied and controlled, an Ishikawa diagram was constructed, see Figure 1. In addition, a risk estimation matrix (REM) was carried out to rank the previously identified analytical conditions, see Table 1 [17,30].

##### Equipment Qualification

Rheometer qualification was performed by determining the viscosity profile of a reference standard. Two temperatures were considered, 25 °C and 32 °C. The first one reported to the standard manufacturer specifications, whilst the second aimed to reproduce the previously reported method conditions.

Triplicate measurements were performed, on three different days, in order to evaluate method precision.

##### Precision

To test precision twelve rheological measurements for each test, were conducted, on three different days in order to comply with the updated EMA requirements. A RSD less than 15% was considered acceptable to validate the previously defined endpoints [6,31].

##### Discriminatory Power

One of the most relevant steps during a validation procedure is the evaluation of the method discriminatory ability, i.e., the capacity of the method to discriminate between different formulations. To address so, the methods sensitivity, specificity and selectivity, should be proven [21,31,32]. 

The rheological profile of F_5_, F_10_ and F_20_ was cross-compared. Furthermore, F_10NC_ rheological profile was determined as an additional discriminatory element. By tracing the rheological profile of such formulations, the discriminatory ability of the method can be sustained, since microstructure differences are highly sensitive to changes in excipient concentration and manufacturing process [5,21,23,33,34,35]. In this context, the sensitivity of the rheological methods was validated by evaluating the CAA response to changes in the concentration of glycerol monostearate. If the CAA obtained with F_5_ were lower than F_10_, and if the F_20_ CAA mean was superior when compared to F_10_, the rheological methods are considered sensitive. 

On the other hand, the specificity of the method was evaluated by assessing whether the considered CAA of F_5_, F_10_ and F_20_ were able to successfully reflect the different glycerol monostearate content. A linear regression model of the CAA as dependent variable by the thickener concentration was used to estimate correlation coefficient (R^2^). The method was considered to be specific if the R^2^ was larger than 0.9 [31,33,36]. The method selectivity was documented statistically. Pairwise comparisons between the reference formulation (F_10_), and the specifically manufactured formulations F_5_, F_20_ and F_10NC_ were conducted. The differences between the means were considered to be significant for values of *p* < 0.05. If the considered CAA of each formulation presented significant differences, the method was considered to be selective. During discriminatory capacity studies, six replicates per formulation, was considered for each rheological measurement. 

##### Robustness

To evaluate the method robustness, the impact of three different experimental setups was assessed. These included temperature fluctuations (+2 °C and −2 °C), sample application (positive displacement syringe vs. spatula) and finally, geometry impact. For rotational studies, the performance of a C35/2°/Ti cone–TMP35 plate (C35-P35) configuration was compared to a P35/Ti plate-TMP35 plate (P35-P35) configuration. For oscillatory measurements, the impact of P35/Ti plate–TMP35 plate (P35-P35) configuration was compared to P20/Ti plate–TMP20 plate (P20-P20) configuration. The method was considered to be robust, if the CAA did not deviate by more than 15% from the mean CAA, at nominal method parameter settings. 

#### 2.2.4. Statistical Analysis

Statistical analysis was performed using GraphPad Prism 5 Software (San Diego, CA, USA) by applying a one-way ANOVA with Tukey multiple comparison test. Differences among mean values were considered statistically significant when *p* < 0.05.

## 3. Results and Discussion

### 3.1. Rheological Method Validation

In the quest of a standardized procedure to assess the rheology profile of topical dosage forms, and underlying the aQbD principles, CMV and CAA were previously identified and their impact crosswise assessed, based on the pillars of method validation. Results are discussed in the sections that follow.

#### 3.1.1. HC Cream Rheological Characterization

As displayed in Figure 2, all formulations exhibited a non-Newtonian and shear thinning behaviour with a consistent decrease in apparent viscosity while increasing the shear stress. The acquired rheograms clearly show three distinct regions: (A) 1st Newtonian range with a plateau value corresponding to the zero-shear viscosity (ƞ_0_); (B) shear-thinning range with shear stress-dependent viscosity function ƞ = f(τ) and (C) 2nd Newtonian range with the plateau value corresponding to the infinite shear viscosity (ƞ_∞_). The ƞ_0_ depicts a formulation viscosity towards an infinitely low-shear rate, close to zero, whereas ƞ_∞_ represents a formulation viscosity towards an infinitely high-shear rate [34,37]. The upper (ƞ_U_) and lower (ƞ_L_) shear-thinning viscosities were also considered. These CAAs encompass initial and final borderline viscosity values of the shear-thinning range.

Formulation viscosity provide a useful information on the release of the active substance from the vehicle. In highly viscous systems, drug release is hampered, affecting its bioavailability and inherent therapeutic effectiveness [2]. Moreover, viscosity results can also shed light on formulation resistance to structure breakdown [38]. Besides the impact on product performance and stability, this CAA also determines formulation appearance, spreadability and retention at the application site, fundamental aspects for patient compliance [39].

The viscosity curves of all formulations displayed a specific yield point (τ_0.ROT_). This CAA was estimated from rotational measurements via tangent crossover method. τ_0_ is known as the minimum shear stress required to initiate material flow or the stress below which a material exhibit gel-like and elastic behaviour. Beyond τ_0_, cream microstructure changes, causing elasticity loss and the flowing of the sample. Formulations with raised τ_0_ consisted on more structured network systems and with increased viscosity, offering higher resistance to external deformation forces [38]. These reasons reinforce the suitability of yield point as a stability indicator CAA. Furthermore, τ_0_ values of pharmaceutical products should be high enough to avoid material flow out of a container when the container is placed in an upside-down position. However, it should not be so large that it offers resistance to flow when spread over the skin [40,41]. Spreadability is a critical sensory property highly dependent on formulations τ_0_ [35]. Thereby, this CAA is likewise an essential element for patient acceptance.

The flow curves (Figure 3) enabled the classification of all formulations as thixotropic systems, since hysteresis loop areas were promptly observed with the rising curves located above the return curves. Thixotropy is a reversible phenomenon exhibited by non-Newtonian materials, characterized by a reduction in the apparent viscosity when the material is subjected to a constant shear rate (deformation), which returns to its viscosity and initial structure when the shear rate is ceased (recovery) [42].

From data analysis, it is possible to observe that formulation glycerol monostearate content produces significant changes in cream microstructure, with flow curves displaying different thixotropic relative areas (S_R_). More structured systems required more time to rebuild the damaged bonds. Such changes are attributed to structure rearrangements that involve rupture and subsequent reformation of weak bonds [43]. Besides the impact on product performance, thixotropy also contributes to an easy formulation spreadability at the application site, fundamental for patient acceptance and compliance [7,44]. Moreover, during shelf-life, cream formulations undergo repeated shear forces when extruded from the container. Hence, to guarantee physical stability against breakdown, microstructure recovery must be ensured through a thixotropic behaviour [34]. For that reason, this CAA is also a good stability indicator.

Regarding the formulation amplitude sweep behaviour (Figure 4), a linear viscoelastic region (LVR) was likewise observed. The LVR is a constant plateau where storage modulus (G′) or loss modulus (G″) values are independent of the strain and only correlated with molecular structure. Within LVR, microstructure is maintained intact and any disruption will be instantaneously recovered [15,45]. All formulations exhibited a well-established yield point (τ_0.OSC_) and flow point (τ_f_) values. Similar to τ_0.ROT_, τ_0.OSC_ is defined as the minimum shear stress that must be applied to induce material flow. However, this CAA corresponds to the shear stress value detected at the end of LVR plateau, obtained through oscillatory measurements.

Beyond τ_0.OSC_, a deviation from LVR is observed with G′ decreasing while G″ simultaneously increasing until τ_f_ [46,47]. τ_f_ is an important rheology property which corresponds to the shear stress value where the modulus crossover (G′ = G″) occurs. τ_f_ can be considered as the borderline between the gel (solid-like) and the fluid (liquid-like) state. Prior to τ_f,_ G′ is higher than G″, suggesting that the system predominantly exhibits elastic properties. Nevertheless, if surpassing this point, the prevalence of G″ over G′ suggests a more viscous system. Any microstructure disturbance after τ_f_ will then produce irreversible deformations in the gel network structure [34].

Rheological data suggested that more structured and viscous formulations offer more resistance to deformation forces, which is denoted by higher LVR plateau, τ_0_ and τ_f_ results [48]. Similar to the τ_0_ response, LVR plateau and τ_f_ are also important stability references.

Regarding frequency sweep profile Figure 5, the four formulations exhibited a dominance of the storage modulus (G′) over the loss modulus (G″). When the material displays a more viscous behaviour, a G′ < G″ tendency is observed; conversely, when the elastic properties of a material prevail, G′ > G″ [49]. Accordingly, the HC cream herein under evaluation can be considered as an essentially viscoelastic system, being its microstructure dominated by a gel network structure [46]. Viscoelastic materials combine two different characteristics: the viscous and the elastic component. The first one, implies that they deform slowly when exposed to external forces (G′ < G″). The term “elastic” implies that once a deforming force has been removed, the material will return to its original structure (G′ > G″) [34]. By other words, during the deformation process, the prevalence of elastic properties also determines a more stable microstructure, since reversible deformations (G′) overcome the irreversible ones (G″) [34]. Besides physical stability, formulation spreadability, drug release and skin bioadhesion, can be impacted by viscoelastic properties [50,51].

Important consideration was also given to loss tangent (tan δ). Tan δ is a dimensionless term that describes the ratio between G″ and G′. This endpoint is useful to elicit information regarding system structure. When tan δ < 1 (G″ < G′), elastic properties and gel-like or solid state dominate; when tan δ > 1 (G″ > G′), viscous properties and a liquid-like or fluid state prevail; when tan δ = 1 (G″ = G′), τ_f_ is achieved [52]. For all the formulations, a tan (δ) close to zero was observed, confirming the gel-like state and elastic properties prevalence. Besides the effect on product performance, those CAAs are important stability indicators with meaningful impact on patient compliance.

#### 3.1.2. Equipment Qualification

Equipment qualification studies were firstly performed to investigate and compare a Newtonian standard flow curve profile to its manufacturer specifications. Viscosity values were provided for two different temperatures −20 °C and 25 °C.

To determine the standard viscosity at 32 °C, Andrade equation can be employed, see Equation (1).
(1)$$\eta =De^{\frac{B}{T}}$$
where D and *B* correspond to empirical constants and *T* to the absolute temperature.

Afterward, the resulting equation provides a close approximation of viscosity as a function of temperature [53,54]. Through this model, it was possible to determine the theoretical viscosity of the standard sample at 32 °C, the selected temperature for this study.

In order to provide a reliable strategy for qualification studies, the standard sample viscosity was also determined at two different temperatures. Firstly, at 25 °C to directly compare with the manufacturer specifications, and then at 32 °C. The obtained viscosity at 32 °C was then cross-compared with the theoretical value calculated through Equation (1).

Acceptance criteria and interday results of equipment qualification studies are summarized in Table 2.

Since the standard presents a Newtonian behaviour, some of the rheological endpoints previously reported for hydrocortisone cream are not applicable. The viscosity of a Newtonian sample is independent of both degree and duration of the applied shear stress, therefore infinite-shear viscosity, lower and upper-shear viscosity all share the same value. Furthermore, according to the same rationale, no yield point is verified [14,34].

Viscosity results at 25 °C comply with the manufacturer specifications. Likewise, the theoretical viscosity at 32 °C, estimated through the Andrade equation, is also compliant with the experimentally determined values. As expected, higher temperatures led to a decrease in viscosity. This is mainly related with an increase of the molecular kinetic energy alongside with the attenuation of weak intermolecular attractions (London dispersion forces). Both occurrences stimulate a molecular realignment in the direction of shear, thus decreasing viscosity [54]. Viscosity results for both temperatures meet the inter-day specification, therefore the equipment proved to be compliant.

#### 3.1.3. Precision

To address the method precision a *n* = 12 was considered during three independent days to evaluate both intraday and interday variability. Results are displayed in Table 3.

The majority of the CAA displayed compliant results concerning both intra and interday evaluations, thus reinforcing the suitability of the proposed methods. Nevertheless, two variables presented high and non-compliant precision results: ƞ_0_ and ƞ_U_.

The main reason that supports this occurrence mainly relates with the non-Newtonian behaviour of the hydrocortisone cream. As previously explained in Section 2.2.1 to warrant a detailed characterization of the flow curve, the acquisition of 3 different segments, 1st Newtonian plateau, shear thinning region and 2nd Newtonian plateau, were actively pursuit. Both ƞ_0_ and ƞ_U_ are retrieved from the first segment of the flow curve. The first endpoint concerns viscosity values at an “infinitely low” shear rate, whilst the second one concerns the viscosity registered prior to the shear thinning behaviour, which occurs at medium shear rates [34].

During the 1st Newtonian plateau, at low shear rates, some sample macromolecules start to lean into a given shear direction. For some of them, this induces partial disentanglements. Consequently, a viscosity decrease is denoted in these parts of the sample. Nevertheless, due to the sample intrinsic viscoelastic behaviour, some other macromolecules, which were already oriented and disentangled, start to recoil and re-entangle all over again, thus inducing a viscosity increase. As a result, during this segment of the viscosity curve, the sum of the partial orientations and re-coilings with the sum of all disentanglements and re-entanglements, results in no significant changes in viscosity [34]. However, these interactions are difficult to replicate, thus explaining the high RSD values, which are not registered in the other rotational endpoints, such as infinite shear viscosity, lower-shear thinning viscosity, rotational yield point and relative thixotropic area. Regarding oscillatory measurements, all the selected CAA demonstrated to be precise in both intraday and interday levels.

Even though the majority of the CAA proved compliance with the previously established criteria (RSD < 15%), which are in agreement with FDA guidelines, a critical evaluation should be made [31]. If the updated EMA criteria (RSD < 10%) was to be followed, three CAA would display non-compliant results (ƞ_L_, S_R_ and τ_0.OSC_) [21].

Similar results were attained by Victor Mangas-Sanjuán and collaborators [6]. The authors performed a comprehensive rheological analysis of 10 different batches of a reference ointment containing calcipotriol and betamethasone. The selected endpoints were: relative thixotropic area, rotational yield stress, zero-shear viscosity, viscosity at 100 s^−1^, loss tangent, elastic and viscous modulus at 1 Hz, and finally m′ and m″ which regard fit and spreadability parameters. The authors evidenced high intra-batch variability in relative thixotropic area and zero-shear viscosity, which were also registered in the present work. Moreover, variability in both elastic and viscous modulus at 1 Hz was also presented. Through different batches comparison, the authors were able to draw several conclusions: (i) some endpoints do not follow a normal distribution and, therefore, do not qualify for comparison according to the EMA criteria; (ii) if a parametric evaluation is performed for low inter-batch variability endpoints EMA criteria can be successfully applied. Nevertheless, endpoints which display high inter-batch variability, equivalence cannot be supported. In conclusion, this work was able to support that a CV of 10% is too strict to conclude equivalence regarding the rheology profile of topical semisolid drug products. In order to promote a practical applicability of the extended pharmaceutical equivalence concept, as desired in the European draft guideline, it is imperative to establish wider criteria based on the inter-batch variability of the product being studied.

#### 3.1.4. Discriminatory Power

A solid documentation of the method discriminatory ability is progressively being demanded by the regulatory authorities, in order to prove that the methods are able to assure a critical distinction among samples.

Even though comprehensive reports addressing the evaluation of this validation component for in vitro release (IVRT) and in vitro permeation methods (IVPT), can be found in the literature, the scenario is slightly different when considering rheology methods [21,26,33,36]. However, since these methods play a central role during semisolid microstructure characterization, the development of a scientific driven platform able to sustain their discriminatory capacity, could be beneficial in regulatory terms. This fact has been extensively discussed in Skin forum (Reims, September 2019) and in EUFEPS Open Forum Discussion on the Draft Guideline on Quality and Equivalence of Topical Products (Bonn, June 2019).

To document the discriminatory power of a method, three concepts should be addressed: sensitivity, specificity and selectivity [31]. For IVRT and IVPT, different strength formulations can be tested to evaluate these concepts. If the methods are able to reflect distinct and proportional in vitro release rate (IVRT) or alternatively, maximal rate of absorption (IVPT), the discriminatory power of both methods is adequately supported. The same rationale can be transposed to rheology methods.

An in-depth formulation knowledge is required to design appropriate and complete validation procedures able to assess the sensitivity, specificity and selectivity. As previously mentioned, two contributions should be mainly accounted for: the impact of the quantitative profile (CMA) and also the influence of critical production parameters (CPP). According to prior knowledge from our group, formulation impact was assessed by varying glycerol monostearate content, since due to its thickening properties, this excipient highly impacts hydrocortisone cream microstructure. Regarding CPP, the homogenization rate proved to be a highly influent CPP and was for this reason selected [11,30].

Discriminatory power results are summarized in Table 4.

Sensitivity evaluation showed that the four rheological methods–CS step test, thixotropy, amplitude and frequency sweep, were able to distinguish the three formulation with different glycerol monostearate content. The reference formulation (F_10_) CAA, presented higher values whenever compared with F_5_ CAA, and as expected, with increasing thickener concentrations (F_20_), all CAA displayed a higher response, see Table 4. For this reason, sensitivity was established.

Rotational and oscillatory methods were also able to successfully establish a linear relationship between thickener concentration and all CAA, thus documenting the method specificity. The determination coefficients for all endpoints were mostly in the range of (0.914–1), indicating a good fitting, see Table 4 and Figure 6.

To evaluate selectivity, the ability of the methods to accurately identify distinct formulations, three pairwise statistical comparisons were performed: (i) F_10_ vs. F_5_; (ii) F_10_ vs. F_20_ and; (iii) F_10_ vs. F_10NC_. The results, summarized in Table 5, demonstrate that for most comparisons low *p*-values are attained, suggesting that there are significant differences among the formulations.

There were, however, non-compliant results observed for tan δ comparison between F_10_ and F_5_ (*p*-value = 0.4165). Notwithstanding, since this parameter regards the ratio between G″ and G′, and these two CAA display significant differences between F_10_- F_5_, this punctual lack of compliance does not undermine overall selectivity results.

#### 3.1.5. Robustness

Method robustness was evaluated by assessing the impact of minor fluctuations in temperature, geometry and sample application. An important outcome of the robustness analysis is to establish appropriate analytical parameters to ensure method validity [55].

Regarding temperature effect, the method is generally robust, however, special attention should be regarded for some CAA which revealed to be more sensitive to this parameter. F_10_ sample testing was conducted at a standard temperature of 32 °C (to mimic skin conditions) and at 30 °C and 34 °C. Under these conditions, a significant decrease on specific CAAs was attained (data not shown), suggesting a disruption on cream microstructure when exposed to rising temperatures. As displayed in Table 6, non-compliant results were accomplished for ƞ_0_, ƞ_U_, S_R_, τ_0.OSC_ and τ_f_ responses, with RSD > 23.18%.

Both rotational and oscillatory measurements are programmed at isothermal conditions, because of temperature effect on structural properties. Depending on excipients glass transition temperature, molecular weight, melting point and molecular rearrangement, a relationship among temperature and rheology CAAs may be established for non-Newtonian systems, since any change on this parameter may produce significant changes on the network structure rigidity and, thus, on product rheology [56].

F_10_ viscosity endpoints tend to decrease with increasing temperatures. Higher temperatures may impact intermolecular forces breakdown mechanisms, deteriorating the network structure and inducing the establishment of less viscous systems [8,34].

F_10_ displays a thixotropic behaviour at all investigated temperatures with a typical hysteresis area. A temperature increase induces smaller S_R_, since low viscous system offer less resistance to deformation forces, requiring a lower shear rate to deform and less time to structure recovery [7,41,57].

The τ_0.OSC_ and τ_f_ values of F_10_ were also highly influenced by temperature. Higher temperatures disrupt intermolecular interactions of the network, resulting in lower τ_0.OSC_ and τ_f_, since a weaker network structure offers low resistance to deformation forces and requires lower shear values to initiate flow and even to structure break [58,59,60]. This is not in agreement with τ_0.ROT_ results, a compliant parameter, suggesting that τ_0_ determination through oscillatory measurements is highly subject to variability.

Considering the geometry impact, this is a critical method variable that requires prior selection and optimization as confirmed by the lack of method robustness.

In the literature, there is no agreement regarding the most suitable geometry configuration for both oscillatory and rotational measurements. In this context, for the selection of an appropriate configuration, sample viscosity, geometry configuration, angle and radius, and gap distance should be carefully considered. Generally, cone-plate configuration is used for bulk liquids and dispersions (suspensions and emulsions) with particle size less than 5 μm, whereas plate-plate configuration is used for dispersions containing relatively large particle size [61].

In this context, geometry impact was assessed in rotational tests pondering distinctive configurations/angles: cone geometry with 2° (C35/2°) and plate geometry with 0° (P35/0°). Note that (data not shown), when comparing P35-P35 configuration with the standard configuration (C35-P35), higher variability results (RSD > 15%) were attained for rotational CAAs intraday measurements. Moreover, as represented in Table 6, non-compliant results were observed for the overall CAAs, with RSD exceeding 20.49%.

The results suggest that cone-plate configuration is preferred to perform rotational measurements, since the shear rate is the same throughout the sample, in opposite to plate-plate configuration, where the shear rate varies along the plate radius, increasing from the center to the edge. The non-uniform shear flow observed in plate-plate configuration seems to produce higher apparent viscosities and likewise an increment on the overall CAAs values. When performing rotational tests, rheology results are influenced significantly by this effect, but it is negligible when performing oscillatory test at small deformation values within the LVR plateau [34,62].

Considering geometry diameter, its selection is mainly dependent on sample viscosity. For low-viscous materials, it is preferable to use a large geometry diameter, therefore, benefiting from a large shear area. Correspondingly, for high-viscous and rigid materials, a smaller dimeter should be selected [63].

In this context, geometry impact was assessed in oscillatory tests considering different plate diameter: 35 mm (P35) and 20 mm (P20). Comparing P20-P20 configuration with the standard configuration (P35-P35) (data not shown), higher variability results (RSD > 15%) were observed for most intraday measurements. As displayed in Table 6, non-compliant results were acquired for τ_0.OSC_, τ_f_, G´ and G″ with an RSD > 19.92%. These results indicate that the plate diameter of 35 mm is the suitable geometry to test cream samples with similar F_10_ viscosity.

As previously mentioned, in a plate-plate (P20-P20) configuration, shear conditions are not uniform along the plate gap and present a high dependence on geometry radius and gap distance. Amplitude sweep test seems to be independent on plate radius. Despite the non-uniform shear conditions provided by this configuration, if measurements are carried out within the LVR plateau, compliance is not compromised. If measurements are performed outside the LVR, higher shear stress values are detected, which result in superior τ_0.OSC_ and τ_f_ values [64,65]. In turn, for frequency sweep measurements, this method variable does not impact G′ and G″ parameters.

In what concerns geometry radius, the obtained higher viscoelastic results may be related with lack of method sensitivity under low shear stress values [34].

Regarding sample application, this is a method variable that should be carefully equated, because of its significant effect on rheology results. In the present study, sample application effect was investigated testing distinct devices: syringe and spatula.

When comparing spatula cream application with syringe cream displacement (data not shown), higher variability results (RSD > 15%) were achieved for CAAs intraday measurements. Even though proved to be compliant for ƞ_∞_, τ_0.ROT_, LVR plateau, G′, G″ and tan δ, rheology method presented higher RSD values for ƞ_0_, ƞ_U_, ƞ_L_, S_R_, τ_0.OSC_ and τ_f_
Table 6.

With respect to rotational measurements, shear thinning range [ƞ_U_–ƞ_L_] is highly influenced by this variable due to the irregularities in sample deposition on the plate.

The non-compliant results for S_R_ confirmed that the syringe device used in this study did not produce sample strain or result in structure loss in contrast with spatula, revealing that this specific CAA is highly influenced by application device. Any non-homogeneity in sample deposition, such as air bubbles, may cause a premature sample rupture and influence the entire rheology profile [61,66]. Furthermore, sample amount (a high impacting variable, see Table 1) is more carefully monitored with a syringe application.

Even though we obtained compliant results for τ_0.ROT_, suggesting that this specific CAA is more robust to application changes, this is not in agreement with τ_0.OSC_ determination through oscillatory measurements, which are non-compliant [34]. τ_f_ determination seems also to be highly sensitive to sample application variability. These results show a strictly dependence on sample application regarding amplitude sweep tests.

#### 3.1.6. Updated Risk Assessment

According to the previously presented results it was possible to update the REM, see Table 7, enlightening the different levels of the main method variables affecting rheology CAAs. The updated levels demonstrate that specific method variables should be carefully pondered due to their significant impact on rheology CAAs.

#### 3.1.7. Standardizing the Procedure

In order to provide a straightforward analysis, the following table summarizes the main outcomes unveiled in this study. In the pursuit of the development and validation of a rheological profile of a semisolid dosage form, the impact of CMVs on specific CAAs should be considered (Table 8).

## 4. Concluding Remarks

In light of the new regulatory requirements, the importance of a detailed rheological characterization of topical semisolid dosage forms is undeniable. A comprehensive framework for the development and validation of the rheology profile acquisition is herein presented. Even though, a 1% *w/w* hydrocortisone cream was used as a case study, the same rationale can be transposed to other semisolid products.

The obtained experimental data revealed that the proposed method is accurate, precise, discriminative and robust. Nevertheless, there are critical method variables that should be optimized prior to experiments. These include geometry, sample application mode and temperature. A broad range of rheological critical analytical attributes were identified: zero-shear viscosity, upper shear thinning viscosity, lower shear thinning viscosity, infinite-shear viscosity, rotational yield point, thixotropic relative area, linear viscoelastic region, oscillatory yield point, storage modulus, loss modulus and loss tangent.

According to the updated risk assessment, the following can be considered as more sensitive monitoring responses: thixotropic relative area, oscillatory yield point and viscosity related endpoints. These rheological attributes are crucial to the formulations physical stability, in vitro performance and, consequently, spreadability and patient compliance.

Moreover, if rheology methods are applied as PAT tool during product manufacture, a close monitoring of the rotational yield point, linear viscoelastic region, storage and loss modulus, as well as loss tangent, can be highly beneficial. The continuous assessment of these parameters enable an early detection of CPP and CMA, responsible for microstructure fluctuations, which in turn would allow a reduction in out of specifications results and overall batch variability of topical dosage forms.

## Figures and Tables

**Figure 1 pharmaceutics-12-00820-f001:**
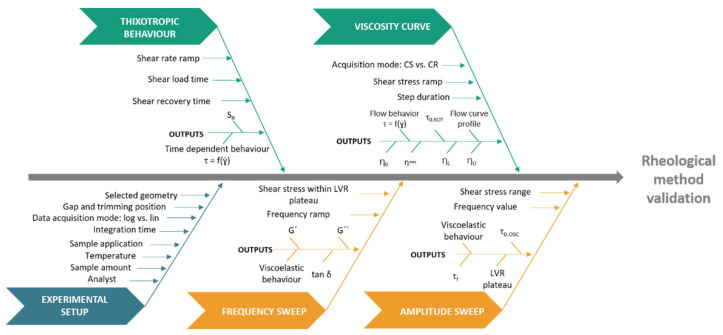
Hypothetical Ishikawa diagram applied to the acquisition and validation of a rheology profile.

**Figure 2 pharmaceutics-12-00820-f002:**
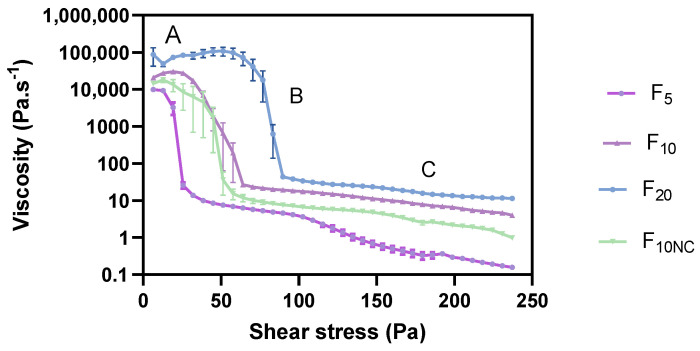
Effect of glycerol monostearate content (F_5_, F_10_ and F_20_) and homogenization rate (F_10NC_) on formulation viscosity curve. (**A**): 1st Newtonian plateau; (**B**): Shear-thinning range; (**C**): 2nd Newtonian plateau. Results report to a 6 < *n* < 12.

**Figure 3 pharmaceutics-12-00820-f003:**
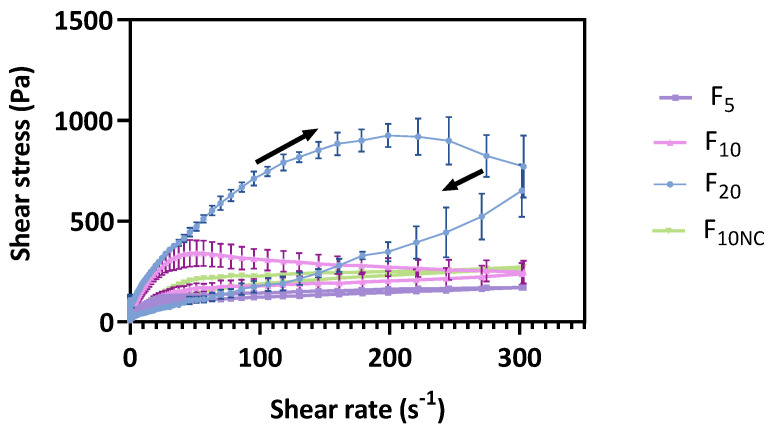
Effect of glycerol monostearate content (F_5_, F_10_ and F_20_) and homogenization rate (F_10NC_) on formulation thixotropic relative area.

**Figure 4 pharmaceutics-12-00820-f004:**
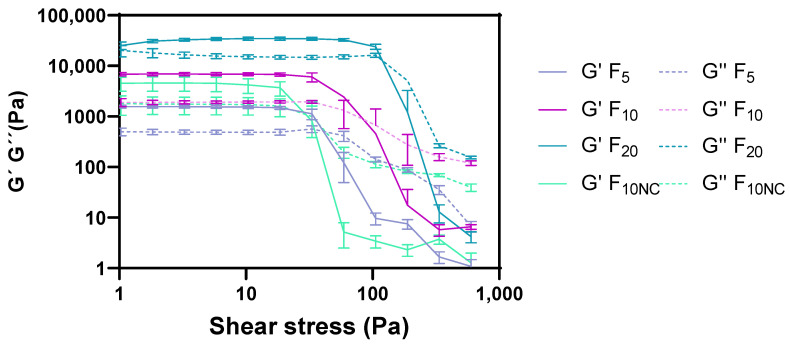
Effect of glycerol monostearate content (F_5_, F_10_ and F_20_) and homogenization rate (F_10NC_) on the formulation amplitude sweep.

**Figure 5 pharmaceutics-12-00820-f005:**
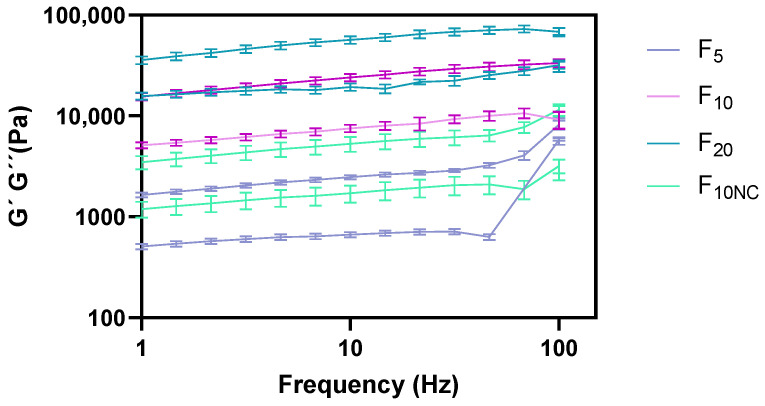
Effect of glycerol monostearate content (F_5_, F_10_ and F_20_) and homogenization rate (F_10NC_) on the formulation frequency sweep.

**Figure 6 pharmaceutics-12-00820-f006:**
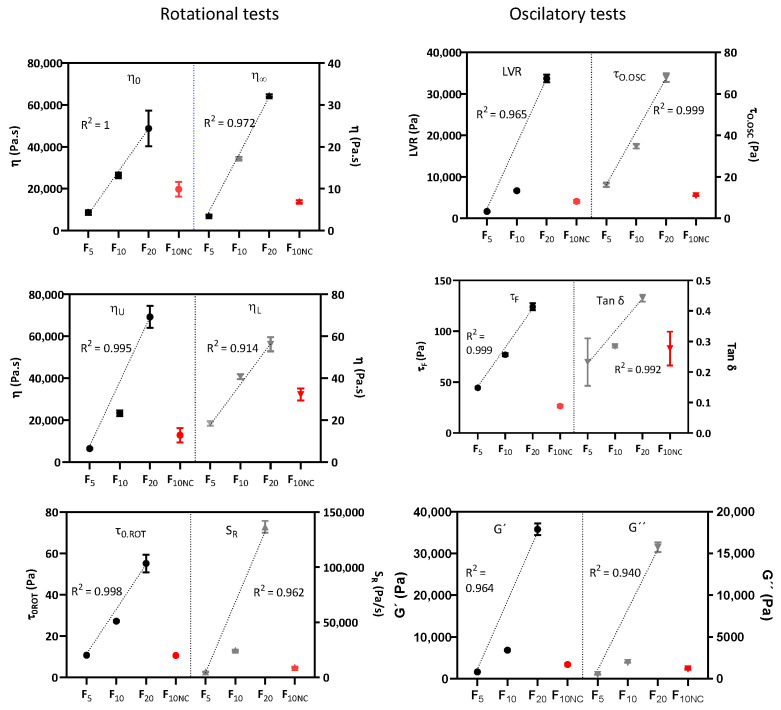
Specificity results. Results are expressed as mean ± SEM. A 6 < *n* < 36 was used.

**Table 1 pharmaceutics-12-00820-t001:** Initial risk estimation matrix (REM) for rheology method validation.

Test	Viscosity Curve					Thixotropic Behaviour	Amplitude Sweep			Frequency Sweep		
CAA	η_0_	η_∞_	η_U_	η_L_	τ_0.ROT_	S_R_	LVR *plateau*	τ_0.OSC_	τ_f_	G′	G″	Tan δ
CMV												
Geometry	M	M	M	M	M	M	M	M	M	M	M	L
Temperature	M	M	M	M	M	M	M	M	M	M	M	L
Sample application	M	M	M	M	M	M	M	M	M	M	M	L
Gap and trimming	L	L	L	L	L	L	L	L	L	L	L	L
Data acquisition mode	L	L	L	L	L	L	L	L	L	L	L	L
Integration time	M	M	M	M	M	M	M	M	M	M	M	L
Sample amount	M	M	M	M	M	M	M	M	M	M	M	L
Analyst	L	L	L	L	L	L	L	L	L	L	L	L
Shear stress ramp	H	H	H	H	H		H	H	H			
Step duration	M	M	M	M	M	M	M	M	M	M	M	L
Shear rate ramp						H						
Shear load time						H						
Shear recovery time						H						
Frequency value							M	M	M			
Shear stress within LVR plateau										H	H	L
Frequency ramp										M	M	L

Key: Low risk variable (Low); Medium risk variable (Medium); High risk variable (High).

**Table 2 pharmaceutics-12-00820-t002:** Predefined acceptance criteria and results for equipment qualification tests. Interday results report to a *n* = 3 performed on three consecutive days. A Newtonian standard with known viscosity was used as reference.

Temperature	CAA	Acceptance Criteria	Results	Status
Standard 25 °C	ƞ (Pa.s)	4.984 Precision (RSD) < 15% Accuracy (Bias) < 15%	Mean = 5.27 ± 0.14 RSD = 2.67% Bias = 5.74%	C
Standard 32 °C		4.360 Precision (RSD) < 15% Accuracy (Bias) < 15%	Mean = 4.8 ± 0.2 RSD = 4.17% Bias = 10.09%	C

Key: shear viscosity (ƞ); Compliant (C), Non-compliant (NC).

**Table 3 pharmaceutics-12-00820-t003:** Acceptance criteria and results of precision evaluation. Results report to a *n* = 12.

CAA	Results					
	Acceptance Criteria	Intraday Variability		Interday Variability		Status
		Mean ± SD	RSD (%)	Mean ± SD	RSD (%)	
ƞ_0_ (Pa.s)	[CAA ± 15%]	26,293 ± 6538	24.87	26,338 ± 7474	28.38	NC
ƞ_∞_ (Pa.s)		17.3 ± 1.5	8.85	17.3 ± 1.6	9.28	C
ƞ_U_ (Pa.s)		23,277 ± 7231	31.06	23,277 ± 8168	35.09	NC
ƞ_L_ (Pa.s)		40.8 ± 5.8	14.14	40.8 ± 6.1	14.83	C
τ_0.ROT_ (Pa)		27.2 ± 1.7	6.36	27.2 ± 1.8	6.71	C
S_R_ (Pa/s)		25,041 ± 2548	10.17	24,576 ± 3238	13.17	C
LVR *plateau* (Pa)		6649 ± 454	6.83	6659 ± 492	7.38	C
τ_0.OSC_ (Pa)		34.6 ± 4.5	13.00	34.7 ± 4.6	13.38	C
τ_f_ (Pa)		76.6 ± 5.3	6.88	76.9 ± 6.4	8.30	C
G′ (Pa)		6867 ± 484	7.05	6853 ±634	9.25	C
G″ (Pa)		1942 ± 148	7.63	1941 ± 184	9.49	C
Tan δ		0.28 ± 0.02	5.57	0.28 ± 0.02	6.87	C

Key: zero-shear viscosity (ƞ_0_); upper shear thinning viscosity (ƞ_U_); lower shear thinning viscosity (ƞ_L_); infinite-shear viscosity (ƞ_∞_); yield point (τ_0.ROT_); relative thixotropic area (S_R_); viscoelastic region (LVR) plateau; yield point (τ_0.OSC_); storage modulus (G′), loss modulus (G″); loss tangent (Tan δ); Compliant (C), Non-compliant (NC).

**Table 4 pharmaceutics-12-00820-t004:** Acceptance criteria and results of discriminatory power evaluation. Results report to mean ± SD. A 6 < *n* < 36 was used.

	Sensitivity						Specificity			Selectivity	
CAA	Results				Acceptance Criteria	Status	Results	Acceptance Criteria	Status	Acceptance Criteria	Status
	F_5_	F_10_	F_20_	F_10NC_							
	Mean ± SD						R^2^				
ƞ_0_ (Pa.s)	8600 ± 2409	26,338 ± 7474	62,870 ± 6630	19,785 ± 6121	CAA [F_5_] < CAA [F_10_] < CAA [F_20_]	C	1.000	R^2^ > 0.9	C	CAA [F_5_] ≠ CAA [F_10_] ≠ CAA [F_20_] ≠ CAA [F_10NC_]	C
η_∞_ (Pa.s)	3.57 ± 0.56	17.3 ± 1.6	17.3 ± 0.4	6.96 ± 0.55		C	0.972		C		C
ƞ_U_ (Pa.s)	6422 ± 553	23,278 ± 8168	69,250 ± 5260	12,815 ± 5969		C	0.995		C		C
ƞ_L_ (Pa.s)	18.4 ± 2.3	40.8 ± 6.1	56.2 ± 3.4	32.2 ± 4.9		C	0.914		C		C
τ_0.ROT_ (Pa)	10.8 ± 0.1	27.2 ± 1.8	55.2 ± 7.4	10.7 ± 0.05		C	0.998		C		C
S_R_ (Pa/s)	5006 ± 325	24,576 ± 3228	136,625 ± 9419	9062 ± 1195		C	0.962		C		C
LVR plateau (Pa)	1636 ± 06	665 ± 491	33,721 ± 2446	4081 ± 900		C	0.965		C		C
τ_0.OSC_ (Pa)	16 ± 2.2	34.7 ± 4.6	67.9 ± 5.4	10.9 ± 0.2		C	0.999		C		C
τ_f_ (Pa)	44.4 ± 1.9	76.9 ± 6.4	124 ± 10	26.4 ± 3.5		C	0.993		C		C
G′ (Pa)	1649 ± 97	6853 ± 634	35,787 ± 634	3419 ± 487		C	0.964		C		C
G″ (Pa)	509 ± 35	1941 ± 184	15,739 ± 184	1169 ± 202		C	0.940		C		C
Tan δ	0.23 ± 0.13	0.28 ± 0.02	0.44 ± 0.02	0.28 ± 0.12		C	0.991		C		C

Key: Compliant (C), Non-compliant (NC).

**Table 5 pharmaceutics-12-00820-t005:** Selectivity results. A one-way analysis of variance (ANOVA) with a Tukey multiple comparison test was performed. The differences between the means were considered to be significant for values of *p* < 0.05.

CAA	F_10_ vs. F_5_	F_10_ vs. F_20_	F_10_ vs. F_10.NC_
ƞ_0_ (Pa.s)	Normal distribution? Yes. CI: [−27,445–8031] *p*-value: < 0.0001	Normal distribution? Yes. CI: [−34,029–10,854] *p*-value: < 0.0001	Normal distribution? Yes. CI: [−5034–18,141] *p*-value: 0.4403
ƞ_∞_ (Pa.s)	Normal distribution? Yes. CI: [−15.49–12.03] *p*-value: < 0.0001	Normal distribution? Yes. CI: [−17.23–12.52] *p*-value: < 0.0001	Normal distribution? Yes. CI: [8.311–12.44] *p*-value: < 0.0001
ƞ_U_ (Pa.s)	Normal distribution? Yes. CI: [−26,774–6936] *p*-value: 0.0003	Normal distribution? Yes. CI: [−61,072–30,873] *p*-value: < 0.0001	Normal distribution? Yes. CI: [−492–21,417] *p*-value: 0.0659
ƞ_L_ (Pa.s)	Normal distribution? Yes. CI: [−29.36–15.52] *p*-value: < 0.0001	Normal distribution? Yes. CI: [−26.67–4.147] *p*-value: 0.0040	Normal distribution? Yes. CI: [0.3611–16.81] *p*-value: 0.0380
τ_0.ROT_ (Pa)	Normal distribution? Yes. CI: [−21.04–11.90] *p*-value: < 0.0001	Normal distribution? Yes. CI: [−31.92–23.91] *p*-value: < 0.0001	Normal distribution? Yes. CI: [12.01–21.15] *p*-value: < 0.0001
S_R_ (Pa/s)	Normal distribution? Yes. CI: [−25,597–13,543] *p*-value: < 0.0001	Normal distribution? Yes. CI: [−118,645–105,452] *p*-value: < 0.0001	Normal distribution? Yes. CI: [9487–21,541] *p*-value: < 0.0001
LVR plateau (Pa)	Normal distribution? Yes. CI: [−6224–3821] *p*-value: < 0.0001	Normal distribution? Yes. CI: [−28,200–25,925] *p*-value: < 0.0001	Normal distribution? Yes. CI: [1376–3779] *p*-value: < 0.0001
τ_0.OSC_ (Pa)	Normal distribution? Yes. CI: [−23.67–13.6] *p*-value: < 0.0001	Normal distribution? Yes. CI: [−37.97–28.41] *p*-value: < 0.0001	Normal distribution? Yes. CI: [18.41–29.11] *p*-value:< 0.0001
τ_f_ (Pa)	Normal distribution? No. CI: [−40.5–24.55] *p*-value: < 0.0001	Normal distribution? No. CI: [−54.46–39.88] *p*-value: < 0.0001	Normal distribution? No. CI: [42.52–58.49] *p*-value: < 0.0001
G′ (Pa)	Normal distribution? Yes. CI: [−6958–3451] *p*-value: < 0.0001	Normal distribution? Yes. CI: [−30,402–27,466] *p*-value: < 0.0001	Normal distribution? Yes. CI: [1966–4902] *p*-value: < 0.0001
G″ (Pa)	Normal distribution? Yes. CI: [−2120–745.8] *p*-value: < 0.0001	Normal distribution? Yes. CI: [−14,373–13,223] *p*-value: < 0.0001	Normal distribution? Yes. CI: [197.6–1348] *p*-value: < 0.0044
Tan δ	Normal distribution? Yes. CI: [−0.1384–0.03657] *p*-value: 0.4165	Normal distribution? Yes. CI: [−0.2315–0.08499] *p*-value: < 0.0001	Normal distribution? Yes. CI: [−0.06649–0.08001] *p*-value: 0.9947

**Table 6 pharmaceutics-12-00820-t006:** Acceptance criteria and results of robustness evaluation for the optimal formulation. Results report to mean ± SD. A *n* = 12 was used.

CAA	TEMPERATURE					GEOMETRY					APPLICATION				
	Acceptance Criteria	Condition	Mean ± SD	RSD (%)	Status	Acceptance Criteria	Condition	Mean ± SD	RSD (%)	Status	Acceptance Criteria	Condition	Mean ± SD	RSD (%)	Status
ƞ_0_ (Pa.s)	[CAA ± 15%]	32 ± 2 °C	25,953 ± 6810	26.24	NC	[CAA ± 15%]	C35-P35 vs. P35-P35	22,981 ± 10,560	45.95	NC	[CAA ± 15%]	Syringe vs. Spatula	26,363 ± 7070	26.82	NC
ƞ_∞_ (Pa.s)			17.2 ± 1.6	9.17	C			15.7 ± 4.4	27.93	NC			17.7 ± 2.0	11.38	C
ƞ_U_ (Pa.s)			22,194 ± 7370	33.21	NC			20,702 ± 10,313	49.82	NC			22,203 ± 8126	36.60	NC
ƞ_L_ (Pa.s)			41 ± 6	14.06	C			37 ± 11	30.17	NC			44 ± 10	22.46	NC
τ_0.ROT_ (Pa)			27.0 ± 2.1	7.74	C			28.4 ± 5.8	20.49	NC			27.3 ± 1.7	6.36	C
S_R_ (Pa/s)			27,602 ± 6397	23.18	NC			30,014 ± 5245	60.53	NC			27,349 ± 8034	29.38	NC
LVR plateau (Pa)			6539 ± 453	8.61	C		P35-P35 vs. P20-P20	6879 ± 996	14.48	C			6704 ± 503	7.51	C
τ_0.OSC_ (Pa)			39 ± 9	25.43	NC			36 ± 7	19.92	NC			38 ± 9	23.31	NC
τ_f_ (Pa)			87 ± 35	26.69	NC			81 ± 17	20.96	NC			85 ± 20	24.03	NC
G′ (Pa)			6783 ± 623	9.18	C			7430 ± 2251	30.30	NC			6961 ± 709	10.18	C
G″ (Pa)			1932 ± 179	9.28	C			2143 ± 712	33.21	NC			1997 ± 263	13.17	C
Tan δ			0.286 ± 0.018	6.44	C			0.288 ± 0.023	7.97	C			0.287 ± 0.021	7.39	C

**Table 7 pharmaceutics-12-00820-t007:** Updated risk estimation matrix (REM) for rheology method validation.

Test	Viscosity Curve					Thixotropic Behaviour	Amplitude Sweep			Frequency Sweep		
CAA	η_0_	η_∞_	η_U_	η_L_	τ_0.ROT_	S_R_	LVR *Plateau*	τ_0.OSC_	τ_f_	G′	G″	Tan δ
CMV												
Temperature	H	L	H	L	L	H	L	H	H	L	L	L
Geometry	H	H	H	H	H	H	L	H	H	H	H	L
Sample application	H	L	H	H	L	H	L	H	H	L	L	L
Gap and trimming	L	L	L	L	L	L	L	L	L	L	L	L
Data acquisition mode	M	M	M	M	M	L	L	L	L	L	L	L
Integration time	M	M	M	M	M	M	M	M	M	M	M	L
Sample amount	M	M	M	M	M	M	M	M	M	M	M	L
Analyst	M	M	M	M	M	M	M	M	M	M	M	L
Shear stress ramp	H	H	H	H	H		H	H	H			
Step duration	M	M	M	M	M	M	M	M	M	M	M	L
Shear rate ramp						H						
Shear load time						H						
Shear recovery time						H						
Frequency value							M	M	M			
Shear stress within LVR plateau										H	H	L
Frequency ramp										M	M	L

Key: Low risk variable (Low); Medium risk variable (Medium); High risk variable (High).

**Table 8 pharmaceutics-12-00820-t008:** Standardizing rheological methodology.

Test	Pre-Setting	CMV	Highly Relevant Caas	Interpretation	Typical Graphical Representation
Rotational: viscosity curve behaviour	Acquisition mode Shear stress ramp Step duration	Temperature: medium risk variable Geometry: high risk variable Sample application: medium risk variable	η_0_ η_U_ η_L_ η_∞_	Higher η_0_, η_U_, η_L_, η_∞_ and τ_0.ROT_ suggest more structured systems	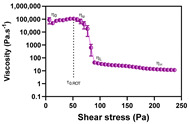
Rotational: shear stress/deformation diagram			τ_0.ROT_		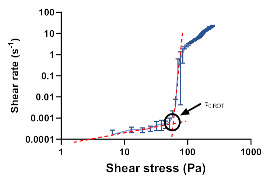
Rotational: thixotropic behaviour	Step duration Shear rate ramp Shear load time Shear recovery time		S_R_	Larger S_R_ is indicative of more structured and consistent (high viscous and less flowable samples) systems	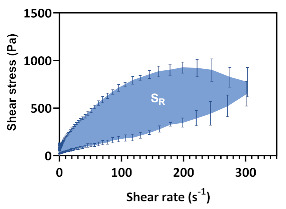
Oscillatory: amplitude sweep.	Frequency value		LVR *plateau* τ_f_	Larger LVR and superior τf are indicative of more structured systems.	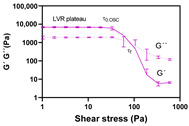
Oscillatory: frequency sweep	Shear stress within LVR plateau Frequency ramp		G′ G″ Tan δ	G′ > G″, prevalence of elastic properties Tan δ < 1, Viscoelastic with prevalence of elastic properties, meaning gel/solid—like structures	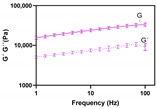

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
