# Peer review of "Rheology by Design: A Regulatory Tutorial for Analytical Method Validation"

_pharmaceutics, 2020, doi:10.3390/pharmaceutics12090820_

Round 1

Reviewer 1 Report

Dear authors,

This manuscript is relevant and the results obtained from validation of the rheology profile acquisition is very important nowadays in order to accomplish the regulatory framework.

The methodologies, the results and the discussion presented in this manuscript they are original, they are well delineated and supported by what I look for in favor of their publication.

Nevertheless, there are mistakes that must be rectified, namely:

  • Pages and line numbers
  • Error! Reference source not found
  • misspelled words, such as formulation

Comments to the authors:

  1. What is the difference between rotational and oscillatory measurements? What is the output of both of these methodologies? Do they affect the microstructure in the same way? These questions should be must be completely clarified before a risk analysis and critical analytical attributes (CAA) is carried out.
  2. The HC Cream Formulations preparation method is detailed but the full composition is missing. Furthermore, the method of preparing of F10NC is missing.
  3. Figure 1: Why is creep mentioned? The output of creep are not the ones mentioned in this figure. Please clarify.
  4. Sample application: A syringe was used and the authors discuss the results obtained with a syringe and a spatula. Does the strength of the syringe not change the structure of the creams? Usually when a syringe is used, usually there is a recovery time before starting the determinations that can be up to 12 h in order to reduce the interference of this parameter. The risk assessment of this parameter is not clear.
  5. Sample amount: This parameter should not be a High risk variable. Sample application presents a much higher risk when compared to the sample amount.
  6. Shear thinning is the correct term of pseudplastic.
  7. What do the authors mean by gel-like and elastic behavior? Gel like=solid like?
  8. Line 307: Beyond τ0, cream microstructure collapses, causing viscosity and elasticity loss . This sentence should be written as follows: Beyond τ0, cream microstructure changes, causing elasticity loss and the flowing of the sample.
  9. Line 328: More structured and viscous systems required more time to rebuild he broken binds. Probably the authors wanted to write: More structured systems required more time to rebuild he damaged bonds.
  10. Line 361: “can be considered as an essentially elastic system, being its microstructure dominated by a gel network structure” . Probably the authors wanted to write: can be considered as an essentially viscoelastic system, being its microstructure dominated by a solid network structure. To be sure of this a creep analysis should be performed.
  11. Line 365-368: Not always, since rheological parameters are not the driving forces of release and permeation. Please rephrased.
  12. Line 372-379: the viscoelastic concept must be introduced.
  13. Table 8: “Larger SR is indicative of more structured and viscous systems”. Probably the authors wanted to write: Larger SR is indicative of more structured and consistent systems”
  14. Larger LVR and superior τf are indicative of more structured and viscous systems. This sentence should be: Larger LVR and superior τf are indicative of more structured systems.
  15. Tan δ < 1, prevalence of elastic properties, meaning and gel - like or solid state. This  sentence should be: Tan δ < 1, Viscoelastic with prevalence of elastic properties, meaning and gel/solid – like structures.  

“Viscous” in the rheology conception is a term of flow. Is does not mean “Consistent”

Author Response

Reviewer #1

  1. Pages and line numbers

R: It was corrected.

  1. Error! Reference source not found

R: It was corrected.

  1. Misspelled words, such as formulation

R: It was corrected.

  1. What is the difference between rotational and oscillatory measurements? What is the output of both of these methodologies? Do they affect the microstructure in the same way? These questions should be must be completely clarified before a risk analysis and critical analytical attributes (CAA) is carried out.

R: The difference between rotational and oscillatory measurements is now provided, please see lines 188-196. The outputs of both methodologies are summarized in lines 201-208, 212-215. 

  1. The HC Cream Formulations preparation method is detailed but the full composition is missing. Furthermore, the method of preparing of F10NC is missing.

R: The full composition of the hydrocortisone cream cannot be disclosed, due to a confidentially agreement. The production method of F10NC was the same used for the other formulations, the solely exception relied on the concentration of glycerol monostearate, a thickener agent. Previous studied by our group reveled that this excipient highly influences cream microstructure, please see Progressing Towards the Sustainable Development of Cream Formulations, Pharmaceutics 2020, 12(7), 647; https://doi.org/10.3390/pharmaceutics12070647. 

  1. Figure 1: Why is creep mentioned? The output of creep are not the ones mentioned in this figure. Please clarify.

R: The authors agree with the reviewer comment. The creep test was erased from Figure 1.

  1. Sample application: A syringe was used and the authors discuss the results obtained with a syringe and a spatula. Does the strength of the syringe not change the structure of the creams? Usually when a syringe is used, usually there is a recovery time before starting the determinations that can be up to 12 h in order to reduce the interference of this parameter. The risk assessment of this parameter is not clear.

R: The authors agree with the reviewer comment. Sample extrusion by a syringe can indeed induce microstructural alterations of the cream. Therefore, the influence of this specific parameter was ranked higher in Table 1.

According to our results, sample application presented itself as a high risk parameter solely for ƞ0, Ƞu, ȠL; SR; τ0.OSC and finally for τf CAA. Please see Table 7.

  1. Sample amount: This parameter should not be a High risk variable. Sample application presents a much higher risk when compared to the sample amount.

R: The authors agree with the reviewer comment. Sample amount should not be ranked as a high risk parameter. Nevertheless, in this study, pivotal studies were firstly performed in order to access what was the optimal amount of formulation for each test. The sample amount was then maintained throughout the study. Therefore, the influence of this specific parameter was ranked as medium in Table 1.

  1. Shear thinning is the correct term of pseudplastic.

R: It was corrected. Please see lines 296.   

  1. What do the authors mean by gel-like and elastic behavior? Gel like=solid like?

R: The authors intended to explain that when the material exhibits a prevalence of G’ > G’’ the elastic properties (gel = solid like) prevail over the viscous (fluid = liquid like) behavior. On the other hand, when G’’ > G the viscous properties overcome the elastic ones. Therefore, during a deformation process, the prevalence of elastic properties determines a more stable structure, since reversible deformations (G`) overlap the irreversible ones (G``). The sentence was rewritten. Please see lines 362-366.   

  1. Line 307: Beyond τ0, cream microstructure collapses, causing viscosity and elasticity loss. This sentence should be written as follows: Beyond τ0, cream microstructure changes, causing elasticity loss and the flowing of the sample.

R: The authors agree with the reviewer comment. The sentence was rewritten as suggested. Please see lines 319-321.   

  1. Line 328: More structured and viscous systems required more time to rebuild he broken binds. Probably the authors wanted to write: More structured systems required more time to rebuild he damaged bonds.

R: The authors agree with the reviewer comment. The sentence was rewritten as suggested. Please see lines 340-342.   

  1. Line 361: “can be considered as an essentially elastic system, being its microstructure dominated by a gel network structure”. Probably the authors wanted to write: can be considered as an essentially viscoelastic system, being its microstructure dominated by a solid network structure. To be sure of this a creep analysis should be performed.

R: The sentence was rewritten as suggested by the reviewer. Please see lines 383-385.

In order to evaluate viscoelastic behavior, creep tests and/or oscillatory tests can be used. In this work, solely oscillatory tests were performed. Oscillatory tests enable an exhaustive characterization of the sample viscoelastic properties, by determining the linear viscoelastic region, the yield pint, flow point, the prevalence of the storage modulus or loss modulus, and loss tangent. All these parameters are interconnected and their relationship is well described in the literature. Creep tests, in industrial practice, have lost of importance since rheometers with air bearings allow the user to directly preset very low rotational speeds, such as the ones used in the present work (0.01 to 250 Pa).

  1. Line 365-368: Not always, since rheological parameters are not the driving forces of release and permeation. Please rephrased.

R: The phrase was rewritten, as suggested by the reviewer. Please lines 390-395.

  1. Line 372-379: the viscoelastic concept must be introduced.

R: The viscoelastic concept was introduced. Please see lines 385-387.

  1. Table 8: “Larger SR is indicative of more structured and viscous systems”. Probably the authors wanted to write: Larger SR is indicative of more structured and consistent systems”

R: The sentence was rewritten. Please see Table 8.

  1. Larger LVR and superior τf are indicative of more structured and viscous systems. This sentence should be: Larger LVR and superior τf are indicative of more structured systems.

R: The authors agree with the reviewer comment. The sentence was rewritten as suggested. Please see Table 8.   

  1. Tan δ < 1, prevalence of elastic properties, meaning and gel - like or solid state. This sentence should be: Tan δ < 1, Viscoelastic with prevalence of elastic properties, meaning and gel/solid – like structures.

R: The authors agree with the reviewer comment. The sentence was rewritten as suggested. Please see Table 8.   

  1. “Viscous” in the rheology conception is a term of flow. Is does not mean “Consistent”

R: Please see answer 16 and Table 8.

Reviewer 2 Report

Minor points:

We suggest to remove the message "Error! Reference source not found" which appear throughout the text instead of the figure or table number.

Author Response

We suggest to remove the message "Error! Reference source not found" which appear throughout the text instead of the figure or table number.

R: It was suitably corrected.